# SIMpLE: A Mobile Cloud-Based System for Health Monitoring of People with ALS [note 1]

**DOI:** 10.3390/s21217239

**Published:** 2021-10-30

**Authors:** Arrigo Palumbo, Nicola Ielpo, Barbara Calabrese, Domenico Corchiola, Remo Garropoli, Vera Gramigna, Giovanni Perri

**Affiliations:** 1Department of Medical and Surgical Sciences, Magna Graecia University of Catanzaro, 88100 Catanzaro, Italy; palumbo@unicz.it (A.P.); ielpon@unicz.it (N.I.); 2Corchiola Computer Science Consulting, 87100 Cosenza, Italy; domenico.corchiola@gmail.com; 3Garropoli Computer Science Consulting, 87100 Cosenza, Italy; remo@garropoli.it; 4Neuroscience Research Center, Magna Graecia University, 88100 Catanzaro, Italy; gramigna@unicz.it; 5Radiological Center Perri-Bilotti, 87100 Cosenza, Italy; giovanniperri@radiologiaperri.it

**Keywords:** neurophysiological monitoring, mHealth, cloud, teleconsulting, ALS, EMG

## Abstract

Adopting telemonitoring services during the pandemic for people affected by chronic disease is fundamental to ensure access to health care services avoiding the risk of COVID-19 infection. Among chronic diseases, Amyotrophic Lateral Sclerosis (ALS), also known as Lou Gehrig’s disease, is a progressive neurodegenerative disease of adulthood, caused by the loss of spinal, bulbar and cortical motor neurons, which leads to paralysis of the voluntary muscles and, also, involves respiratory ones. Therefore, remote monitoring and teleconsulting are essential services for ALS patients with limited mobility, as the disease progresses, and for those living far from ALS centres and hospitals. In addition, the COVID 19 pandemic has increased the need to remotely provide the best care to patients, avoiding infection during ALS centre visits. The paper illustrates an innovative, secure medical monitoring and teleconsultation mobile cloud-based system for disabled people, such as those with ALS (Amyotrophic Lateral Sclerosis). The design aims to remotely monitor biosignals, such as ECG (electrocardiographic) and EMG (electromyographic) signals of ALS patients in order to prevent complications related to the pathology.

## 1. Introduction

The coronavirus disease-19 (COVID-19) epidemic is a public health emergency of international concern. Telemedicine is an effective option with which to fight the outbreak of COVID-19 [1,2] since it provides health services through information technologies and the data network, allowing them to function at a distance. In addition, thanks to telemedicine, it is possible to provide healthcare assistance to patients who cannot access it directly for various reasons [3,4,5,6]. Telecommunication systems make it possible to transmit medical information in the form of texts, images and sounds for the prevention, diagnosis, treatment, and remote control of the patient, avoiding physical movement. Thus, telemedicine allows people affected by chronic diseases to receive adequate health care services [7,8,9]. In addition, telemedicine allows out-of-hospital control of patients suffering from chronic diseases, ensuring timely intervention in emergencies [10,11,12,13,14].

Among chronic diseases, Amyotrophic Lateral Sclerosis (ALS) is a progressive neurodegenerative disease [15]. It is characterized by the progressive degeneration of the I and II motor neurons, nerve cells responsible for voluntary movement; progressive muscle paralysis occurs involving the ability to move, speak, swallow, and breathe [16]. At the moment, there are no therapies able to cure the disease or stop its course. In most cases, survival is around 2–4 years, while only a tiny percentage of patients have a disease of duration more than ten years. Assistance with ALS requires a multidisciplinary approach with a global care path involving the patient, family members, and caregivers. The progressive loss of autonomy makes it necessary to plan support interventions and use advanced technologies. For example, the person with ALS needs not only to move (mobility aids) and to communicate feelings and needs (alternative augmentative communication) but also to ensure vital functions such as breathing and swallowing. Therefore, careful follow-up of the patient’s condition is needed to decide on appropriate interventions. Multidisciplinary teams follow people with ALS in ALS centres. Remote monitoring and teleconsulting could be particularly relevant services for ALS and other chronic disease patients [17]. In addition, the COVID 19 pandemic has increased the need to remotely provide the best care to patients, avoiding infection during ALS centre visits [18,19].

Currently, telemedicine services aimed at ALS patients concern the online administration of the ALSFRSr (ALS Functional Rating Scale, revised), which is a tool for assessing the progression of the disease [20,21]. In addition, an Android app is presented that can administer a series of questions about their condition to the patient [22]. Neurological examinations, which represent the other important tool for assessing the progression of the disease, have been proposed via videoconferencing and have shown promising results and satisfaction on the part of the patients [23,24,25]. However, they have some limitations because it is difficult for the doctor to assess muscle tone, for example. A feasibility study of a telemonitoring system was proposed in [26], in which the authors try to use wearable sensors to evaluate HRV (Heart Rate Variablity) and motor activity of ALS patients. The work does not present the design of a complete system but is only a feasibility study. Therefore, to the best of our knowledge, no systems offer a complete and reliable solution that integrates biosignals telemonitoring and teleconsultation services for ALS patients.

This paper aims to present the design and implementation of an innovative medical mobile cloud-based system, named SIMpLE, to improve the monitoring of disease complications in patients affected by neurodegenerative diseases, such as ALS patients, and the elderly. Cloud computing could achieve better telemedicine services, thanks to on-demand access anywhere, anytime, low costs, and high elasticity [27]. In the last decade, many studies and projects presented cloud-based systems, especially for telecardiology and remote monitoring of vital signs [28,29,30]. The SIMpLE system aims to ensure a better quality of life and promote greater social inclusion for all people with severe disabilities.

The SIMpLE system is a platform for remote monitoring used to acquire and process biosignals, such as ECG and EMG signals of ALS patients. The SIMpLE system monitors vital physiological user parameters and biosignals in real-time to ensure high-security conditions. Furthermore, the platform allows medical staff to monitor the patient’s health conditions at a distance, allowing them to remotely intervene directly on the electronic instrumentation. A prototype of the SIMpLE system was presented in [31], in which the authors illustrated the architecture and the functionalities of the web-based system. In the current paper, we describe the complete system. Specifically, we focus on the mobile application that collects physiological data and transmits them to the cloud system and on the design of the mobile version of the web-based system. The paper is organized in the following way: in Section 2, a general overview of the system’s architecture is given. Then, Section 3 presents the mobile system which aims to collect and transmit biosignals; Section 4 illustrates the cloud-based system for data storage, analysis and teleconsultation. Section 5 describes the functionalities of the mobile application. Finally, Section 6 discusses the main results and concludes the paper.

## 2. Simple System Overview

The hardware and software components of the SIMpLE system are presented in this section. Figure 1 shows an overview of the proposed system.

The SIMpLE modules (Figure 2) are (i) sensors nodes to acquire physiological signals (i.e., EMG and ECG signals) and other vital parameters, (ii) SIMpLE mobile, a sort of hub through which the acquired signals are sent to the web-based system; (iii) the SIMpLE cloud-based system. The latter includes: the patient summary management services (personal data, pathologies, clinical exams), the remote control, the visualization and the analysis of acquired signals, the management of diagnostic imaging conducted on the patient and in compliance with the DICOM standard, the teleconsultation subsystem.

The SIMpLE Mobile module consists of a Raspberry PI4 with an Ubuntu server 20.04.1 operating system. An application created in C♯ with the framework .NET 5, Microsoft cross-platform technology, listening on Bluetooth, receives data from the acquisition sensor nodes. The sensor nodes are Bluetooth sensors for biopotentials and other physical parameters. Through pre-processing of the acquired data, the device will manage the data communication to the Cloud system reliably, for example by managing queues and re-transmissions in the event of an internet connection fault. The data are sent to the server through RESTful Web API calls. The received information is persistent in the case of system crashes and lack of internet connection. Persistence is achieved through a local SQL database. When normal operating conditions are restored, the data are retransmitted.

The SIMpLE application comes with three client applications:An application developed with .NET Core 5 to transfer sensor data;A PHP web application;A mobile application developed with Xamarin Forms which supports iOS, Android, and the Universal Windows Platform (UWP).

## 3. Simple Mobile System: Design and Implementation

The Raspberry Pi4 has been configured as a WiFi access point. It is possible to connect it with a tablet with the SIMpLE mobile app on board, described in details in Section 5. It is possible in safety conditions to configure the system and check the parameters or check the Bluetooth or internet connection. The Raspberry Pi 4 device was selected because it is a low-cost platform with good performance and is suitable for installation of the Linux operating system. Moreover, the Raspberry Pi 4 has built-in WiFi and Bluetooth, making it useful for our purposes, even at low cost.

The data arrive at the gateway via Bluetooth controlled by the listening application, created using .NET Core 5 to manage the system. The app functionalities are listed in the as follows:Pairing of Bluetooth devices;Hardware system verification;Sensor verification;Configuration of patient data;Check connectivity;Geo-referencing configuration;

The SIMpLE application includes the following backend services:A patient microservice is a data-driven create, read, update, delete (CRUD) service that consumes an SQL database.A microservice for data acquisition.

These backend services are referred to as the SIMpLE reference application. Client applications communicate with backend services through a Representational State Transfer (REST) web interface.

The first designed software object is a TCP/IP server (see Figure 3).

The client prepares an object consisting of an action executed by the server and parameters (argument).



   public class HubCommand

{

    public  ServerCommands Action { get; set; }

    public  object Argument { get; set; }

 }



The commands for acquisition from generic Bluetooth sensors are shown below.



  public enum ServerCommands {

    BTPairedDevices,

    BTRegisterDevicesOnHub,

    Disconnect,

    SendSensorAcquisition,

    SensorConnect

    SensorDisconnect

    SensorStartStream

    SensorStopStream

    SensorStartPublishDataMqTT

    SensorStopPublishDataMqTT

    SensorStartPublishDataCloud

    SensorStopPublishDataCloud

    ShutDown,

    Reboot,

 }



This object will be serialized as a string in JSON format and sent to the server, deserializing the string and executing the command in a thread.



 String[] hubCommandsSerialized = message.Split(’|’);

      string[] hubCommandsSerializedDistinct =

      hubCommandsSerialized.Distinct().ToArray();

   foreach (var cmd in hubCommandsSerializedDistinct)

   {

        if (cmd != "")

{

   Thread thread = new Thread(() => ProcessCommand(cmd.Trim(’|’)));

   thread.IsBackground = true;

   thread.Start();

 }

 }



At the end of the command execution, the server sends an object to the client.



  public class HubResponse {

    public ServerACKEnumeration Ack { get; set; }

    public object Argument { get; set; }

    public object Data { get; set; }

}



Below the types of responses are reported:



 public enum ServerACKEnumeration {

   OKConnection,

   Disconnected,

   NotReacheable,

   SensorConnected,

   SensorConnecting,

   SensorDisconnected,

   SensorStreaming,

   SensorNoStreaming,

   SensorData,

   SensorsList

}



The following commands have been implemented for each type of Bluetooth sensor:Client connection;List of sensors coupled to the hub;Connection to a Bluetooth sensor;Disconnection from a Bluetooth sensor;Start streaming of a Bluetooth sensor;Stop streaming of a Bluetooth sensor;Sending the streaming of an acquisition of the gyroscope to the device in real-time.

At the same time as the streaming starts, the data are saved in a local database and are ready to be sent to the Cloud. SQLite does not need a server; it performs and works on files suitable for the IoT (Figure 4).

The Figure 5 schematically represents the communication between the different modules.

A sensor management class has been implemented for each sensor, having, for example, the following form: BTnamesensorManager. For example, this class could be BTShimmersManager.cs if we consider the Shimmer sensors (http://www.shimmersensing.com, accessed on 17 October 2021). To generalize the speech to any Bluetooth sensor, in addition to tests with Shimmer sensors, a BWT61CL sensor (Wit motion Bluetooth Gyroscope (http://wiki.wit-motion.com, accessed on 17 October 2021) was used. The module integrates a high precision gyroscope and a geomagnetic field sensor and outputs information relating to accelerations (x, y, z), gyroscope (x, y, z) and angles (x, y, z). In addition, a library (API) was created for each sensor to access the sensor’s essential functions. Upon the occurrence of certain conditions or requests, a Thread will send the data to a cloud service for the acquisition, even if deferred over time.



Thread thread = new Thread(() => ServerMonitor());

thread.IsBackground = true;

thread.Start();



It will be possible to connect to the hub (to the TCP/IP server) only via VPN and visualize the list of Bluetooth sensors paired to the hub (see Figure 6). If the sensors are disconnected, they can be reconnected and start streaming after connection. The list of connected devices is saved on the DB of the hub, but it is possible to rescan (slow operation as the serial ports of the hub are checked). If the user starts the stream, it is possible to view the raw data in real-time. Of course, these data can be used on mobile devices for any real-time processing. During the acquisition, the records are saved on the hub database to be sent automatically or manually to the Cloud at any time.

### System Security

Cybersecurity is a critical issue in mobile healthcare applications. The system is hardened and secured according to cybersecurity best practices: firewall, anti-DDoS, SELinux.

SELinux mainly consists of a series of enhancements related to the security of the Linux system, implemented using a mandatory access control architecture, incorporated into the major subsystems of the kernel. It runs directly in the kernel, thus implementing a Mandatory Access Control system. In SELinux mechanisms capable of separating information are provided, precisely based on the requirements of confidentiality and data integrity, two of the fundamental aspects of what is defined as computer security. The configuration and administration of SElinux is a highly complicated activity. To configure Selinux, the operating system kernel was recompiled.

Features of SELinux include:Well-defined security policies and the separation between security policies and applications;Support for applications that query security policy and access control;Independence between the different security policies and formats and security contents;Checks for kernel services;Keeping system integrity protection and data confidentiality separate (security (multilevel));Controls on process initialization and program execution;An open file-system, directory, file and (file-descriptor) checks;Checks on sockets, messages and network interfaces;Control of cached information via the AVC (Access Vector Cache);

An in-depth study was devoted to the firewall and how to mitigate DDOS attacks. In the following paragraph, some details about these security mechanisms are described. Iptables is a powerful firewall built into the Linux kernel and is a part of the Netfilter project. The DoS, i.e., Denial of Service, is an action whose goal is to flood the resources of a computer system that provides a particular service to connected computers. Iptables can be used to filter specific packets, block sources or destination ports and IP addresses, forward packets via NAT, and many other things. It is most commonly used to block destination ports and source IP addresses. When used correctly, iptables is a potent tool that can stop many different types of DDoS attacks. A series of rules have been studied that allows the system to be made robust to attacks.

A VPN (Virtual Private Network) allows the creation of a virtual private network that guarantees privacy, anonymity and data security through a reserved communication channel between devices that do not necessarily have to be connected to the same LAN. For example, the configuration app can be connected to the Raspberry access point and, through an OpenVPN connection, it will safely access the configuration functions. The certificate is issued only to authorized persons.

## 4. Simple Cloud-Based System: Design and Implementation

SIMpLE cloud-based system (available online at the following link: https://biomed.arrigopalumbo.com/progetto-simple/, accessed on 17 October 2021) consists of different subsystems that interact and communicate with the SIMpLE mobile system. The list of subsystems of the cloud-based system consists of the:Web server: the system exposes the interfacing features with the SIMPLE mobile system through web services and the web application to display user-side functions.FTP file server: used to store the streams of data collected during the patient’s examination phase, both on the ECG and EMG sides.Database server: used for storing data, patient summary, and other specific data relating to sensors.DICOM server: intended primarily to store DICOM images extracted from diagnostic imaging performed using investigation tools with techniques of Ultrasound, Computed Bone Mineralometry (MOC), Radiography, Nuclear Magnetic Resonance (MRI), Computed Tomography (CT) and other methods. In addition, this server exposes the functionality of analysis and reporting of investigations on DICOM standards carried out on the patient without further installation of specific software.Teleconsultation interface: it allows each patient to be associated with one or more rooms for teleconsultation with doctors and/or technicians. There is also a generic room to hold general consultation sessions between doctors, technicians and patients.

To develop the web-based software system, technologies used are PHP as a server-side language, Ajax/JQuery for managing asynchronous client-side requests, Bootstrap for formatting and defining responsive layouts, CSS3, HTML5.

### 4.1. Functionalities and User Roles

This paragraph describes the functionalities of the cloud-based storage and teleconsultation system (Figure 7).

Different users can access the platform with different roles and privileges in terms of data and the complete list of features. The system permits the management of user roles and displays for each user only the functions for which he/she is enabled. For example, the patient-user can enable one or more doctors to report on his/her exams; the doctor can upload and analyse the patient’s exams with the analysis tools. On the other hand, technician users can arrange examinations for the patient but cannot report.

User management allows all the C.R.U.D. (Create, Read, Update, and Delete) functions based on the “user”, “user_has_role” and “role” data structures. To add a new user, click on the “ADD USER” button located at the top right of each tab (see Figure 8).

Each user can have one or more roles (e.g., the same user could be registered as an administrator and as a doctor). However, the system does not allow the insertion of two users with the same personal data, managing redundancies and duplication. The users who can access the platform are those who have the status set to “Enabled”. Inserting a user without roles is not allowed.

Moreover, SIMpLE allows geo-localization of the patient to conduct possible geographical analyzes on the data. Thus, it is possible to create and manage a patient summary connected to the patient during his/her entire clinical life. User geo-referencing is not mandatory but desirable, as it could be helpful, especially for patients, for subsequent statistical analyzes. In addition, patient data anonymization has been implemented.

### 4.2. Interfaces

After user access to the system, the interface shows two macro sections (see Figure 7). First, the left frame offers the main menu, which consists of the following functions:Dashboard;User management;Pathology management;Types of acquisitions and clinical exams;Patient Management.

In the right frame, the operating interface is dynamically shown for each function selected from the menu on the left.

The dashboard interface offers an operational interface that allows the combined search of registered patients. Users can use filters to:Create a list of patients for whom a pathology has been diagnosed.Obtain a list of patients followed by a particular doctor or by a specific technician.Obtain a list of patients who have undergone a particular acquisition.

After choosing the combination of search filters, the system searches the database and displays the list of patients corresponding to the selected criteria. A contextual search engine performs a full-text search within the queried data structure in each table within the project, automatically showing the entered results.

It is possible to register and manage all the clinical examinations that the patient can carry out in the system. In the patient file, all the clinical information is acquired and reported under observation, from his/her data to the doctors and technicians authorized to operate on the patient. In addition, it is possible to access the clinical examinations conducted on the patient (see Figure 9).

#### Biosignals and Bioimage Web Viewers

Biomedical signals uploaded to the FTP server can be visualized and analysed with a particular web viewer developed for the project (see Figure 10). The viewer parses the sample stream: samples acquired every millisecond have been displayed and analysed to extract functional clinical parameters. Several valuable parameters in the time and frequency domain can be extracted from signals thanks to the implementation of analysis tools [32]. These tools have been developed in LabVIEW (National Instruments, Austin, TX, USA).

The software architecture implements a subsystem for the storage, analysis and reporting of image files acquired during image diagnostics, such as Ultrasound, Computerized Bone Mineralometry (MOC), Radiography, Nuclear Magnetic Resonance (MRI) and Computed Axial Tomography (CT). In the diagnostic imaging management subsystem of examinations carried out on the patient, various functionalities are displayed, exposed entirely via the web, without the aid of plugin installations and with any browser, including mobile.

The tool for viewing and reporting diagnostic imaging tests conducted on the patient allows multiple simultaneous users to view the exam and, in particular, to report it simultaneously. This practice considerably broadens the spectrum of possible actions linked to shared multi-reporting.

## 5. Simple App

A SIMpLE app for Windows, Android and IOS has been implemented. It interfaces with the sensors and the cloud-based system. The application has been optimized for tablet devices. The graphical interface (GUI) has been unified on the three platforms (Windows, Android, IOS) to have the same user experience. The app allows the same main functions of the web system listed in the previous section, such as the (i) Dashboard; (ii) User management; (iii) Pathology management; (iv) Types of acquisitions and clinical exams and (v) Patient Management.

### 5.1. Simple App: Design and Implementation

This section describes the architecture and functions of the app that will have to communicate with the Cloud both in reading and in writing to allow remote users (Technicians, Doctors and Patients) to interact with the online system.

App–Cloud communication is a fundamental requirement in compliance with the rules relating to roles, including remote ones, established during the design phase of the CLOUD system for storing, reporting and viewing the data entered and relating it to the patient’s file (Patient Summary). The cloud-based web app allows for various features, of which a subset must be exposed for use via the mobile app. The authentication mechanism of the app towards the Cloud will have to guarantee the security of communications, the possibility of creating a sort of session. However, the mobile app is not run on the server but works with the mechanism of RPC simulated through web services, the need to assign the role and relative privileges to the data to the user who logs in. Although there are four roles in the web-based system (Admin, Doctor, Technician and Patient) for the app, we will only allow access to Doctors, Technicians and Patients.

The web service that exports the login and user recognition mechanism from the app consists of two authentication phases:Phase 1: authentication of the app to the Cloud. In this phase, it withdraws a session Token generated by the web service at the following address:$ BASE_PATH / getToken.phpThe above web service receives no input parameters and returns a response in JSON format:{‘‘Token’’: ‘‘Value_of_token’’, ‘‘Salt’’: ‘‘Value_of_salt’’}These values will be helpful for the app to continue with the actual authentication, referred to in Phase 2.Phase 2: in this phase, the app queries the Cloud by passing in input Token and Salt data received during Phase 1 and is returned access if the user is registered in the Web-based system, dynamically generating a new session Token which will be used for any other query by the user to the Cloud.

The web service that checks the access credentials is exposed at the following address:


$ BASE_PATH / dologin.php


Input parameters passed in POST mode to the web service:Salt: must correspond to the value received during phase 1;Token: must correspond to the value received during phase 1;UserPassword: e-mail with which the user is registered;UserLogin: password associated with the user within the web-based system.

The web service always returns a JSON whose structure can change depending on the verification carried out:JSON with only one field in case of error containing only the type of error;JSON with different fields in case of successful authentication.

If the authentication is successful, the web service will return the following JSON:



     {‘‘ResponseMessage’’: ‘‘[SUCCESS] User recognized!’’,

‘‘User id’’: ‘‘User_id_value’’,

‘‘Role’’: ‘‘[ADMINISTRATOR | PATIENT | DOCTORS | TECHNICIAN] ’’,

‘‘ Surname ’’:‘‘ User’s surname ’’,

‘‘Name’’: ‘‘User name’’,

‘‘Email’’: ‘‘User email’’,

‘‘Token’’: ‘‘New unique session token associated with the user.’’

   }



Of particular importance during the whole life cycle of the session will be this session Token returned from phase 2, necessary for any other query described in the rest of the document. This Token will be a random value always generated again for each authentication. Therefore, it will be necessary to store it only for the duration of the app connection session.

### 5.2. Simple App Interface

The user must log in; for all platforms, the login screen is shown in Figure 11.

From the left side menu, it is possible to access the list of patients (see Figure 12 on a paginated grid with a pull-down technique, i.e., the data will be loaded (10 records at a time) as the grid is pulled down and added to the view so as not to overload the server. In this operation, the filters set in the search will be applied.

The patient’s main data modification form is accessed by clicking on the “Open Summary” button.

In the patient summary, it is possible to view the associated pathologies. By clicking on the “Add Pathology” button, it is possible to upload a new pathology to the system via a popup. In addition, doctors can view the exams and acquisitions on the patient summary. The Figure 13 shows the DICOM viewer.

## 6. Discussion

The follow-up of ALS patients has been suspended or limited to reduce the risk of infections due to the pandemic. Telemedicine solutions increase healthcare services access, saving time and cost. Nevertheless, stringent constraints in the adoption and diffusion of these solutions are cost, size, unobtrusiveness, ease of use and network capabilities.

This paper proposes a novel telemonitoring and teleconsultant system for ALS patient monitoring and teleconsultation. Patients such as ALS sufferers experience a rapid worsening of their condition in a short time and therefore require constant medical and psychological support. Therefore, there is the need to design and realize new telemedicine services to avoid a more significant decline in physical and psychological functions. Our system guarantees that the patient does not feel lost and, above all, it allows prevention of complications related to the pathology thanks to continuous and constant monitoring. On the one hand, the mobile cloud-based software allows the patient to be followed by several specialists simultaneously and, on the other hand, extract predictive and historical statistics useful for analysis. The web-based software is a unified system that would allow analysis and reporting, teleconsultation, and the shared management/reporting of the patient’s clinical data on a single management card called the Patient Summary.

Our system is unobtrusive since non-invasive wireless devices are used for the measurement of biopotentials and other physiological parameters.

It has overcome some limits of existing telemonitoring solutions that require intensive training of patients, caregivers and physicians. It is user friendly and easy to use, so it does not require specific abilities for its use. The system is usable on all devices, and it is cross-platform. The interfaces are contextual: after selecting a patient, the doctor can manage all information associated with each patient, such as clinical examinations, biosignals, images and other personal information. SIMpLE does not require the downloading and installation of specific packages or tools, even for signal and image analysis.

Another innovative feature of our system is the implementation of security mechanisms to guarantee patients’ data security and privacy, which is the main bottleneck for the adoption of mobile and cloud technologies in the medical context.

In the coming paragraphs, we will focus on the discussion of two important aspects: usability and user experience and compliance with European regulations for medical devices.

### 6.1. Usability and User Experience

To validate our mobile cloud-based health monitoring system, two aspects have been taken into consideration: usability requirements, considered as a main factor to demonstrate the quality of any software system, and user satisfaction, listed as the principal factor that gives a perception of the software system’s success or failure [33]. Since some of the existing cloud computing environments appear very complex and difficult to use, the scientific literature is moving towards the development of methods and mechanisms to evaluate if the cloud computing environment is usable or not [33]. The easy use of the environment is the main objective to ensure that the product achieves high levels of satisfaction for the user. For this reason, involving the user in the evaluation process is a necessary step to take feedback from and to ascertain if the site will satisfy the user or not. User testing, the best way to understand how real users experience the implemented environment or application, requires that the users test the functionalities and the ease of use of the SIMpLE cloud-based system and the related App and provide feedback to evidence the usability of this environment type. From the users’ testing evaluation, the following emerged:

Concerning design principles:Each page/macro section/functionality of system has a clear purpose and fulfils a specific need for environment users in the most effective possible way;The users, while using the system, obtain information quickly and clearly and the content thereof is easy to read and understand;The colours, which play a key role in influencing user satisfaction, are balanced and harmonious with each other, and contrasting colours are used for the text and background to make it easier to read; in addition, the typeface and font size have been chosen for ease of online reading.The system navigation is effective and intuitive thanks to the use of bread crumbs and the design of clickable buttons, following the ‘three-click rule’ which means users will be able to find the information they are looking for within three clicks [33];The user’s private information is protected in respect of privacy criteria.

Concerning usability characteristics:In terms of usability, user testing confirmed that our system has the following characteristics: effectiveness (time to learn and retention), flexibility (easy navigation process, understandable site direction and an easy ’back’ option), memorability, accessibility, error tolerance, and responsiveness [33].

The extension of the usability test to a larger number of users, planned for the future, could provide further information on additional features and improvements.

### 6.2. Issues on Medical Software Certification

Technological development has allowed the emergence of software capable of contributing to healthcare transformation, for example by enabling doctors to make a diagnosis, helping patients manage their diseases better, and allowing an efficient and immediate exchange of vital information. Consequently, the legislation relating to medical devices has addressed the Regulation of Software used in the medical field in an increasingly explicit, direct and detailed way than previously established. The introduction of the EU Medical Device Regulation (MDR) [34] has placed stringent restrictions and heightened requirements on medical device manufacturers to improve device safety and performance. Interestingly, the MDR guidelines have also been extended to digital health technologies and medical applications, considering them Medical Device Software (MDSW). As a result, medical software manufacturers must now carefully consider the new regulatory requirements adopted by the European Parliament and Council in May 2017.

The term software as a medical device is defined by the Medical Device Coordination Group (MDCG) as a set of instructions that processes input data and creates output data. Medical device software is intended to drive the use of a (hardware) medical device, used alone or in combination. Software must have a medical purpose to be defined as an MDSW, such as diagnosis, prevention, monitoring, treatment, or alleviation of a disease. Specifically, medical software includes:Software that directly controls a medical device (hardware), providing immediate information intended to be used by healthcare professionals or patients (e.g., blood glucose meter software).Software that provides support for physicians (e.g., EEG analysis software).Software intended to process, create, or modify medical information when the software is governed by an intended medical purpose (e.g., searching images for decisions that support a clinical hypothesis as to the diagnosis).Independent software, with an intended medical purpose.

Software may be qualified as MDSW regardless of its location (e.g., operating in the cloud, on a computer, on a mobile phone, or as an additional functionality on a hardware medical device).

Some harmonized standards and the MDCG guide applicability to doctors are listed, focusing on those helpful in satisfying the provisions in the regulation (EU) 2017/745 relating to software qualifying as medical devices in their own right.

EN 62304: Medical Device Software—Software life-cycle processes: provides a framework for the life cycle processes of a software medical device, defining, in particular, the activities and tasks necessary for the development and maintenance of highly reliable and safe software;EN 82304-1:2017 Health Software—Part 1: General requirements for product safety: its purpose is the safety and security of health software designed to work on any IT platform (e.g., fixed or mobile physical devices, virtual machines, the cloud) and intended to be placed on the market without dedicated hardware.EN 62366-1:2015 Medical Devices—Part 1: Application of usability engineering to medical devices: this standard focuses on the application of usability engineering principles to all medical devices in order to optimize the usability aspects related to the safety of patients and all other users.EN ISO 13606 Health informatics—Electronic health record communication:the main purpose is to define a rigorous and stable information architecture that can be used for the communication of part or all of the electronic medical records (EHR) of one or more patients between different EHR systems or between an EHR system and a centralized system of data archives.EN ISO 13485 Medical devices—Quality management systems—Requirements for regulatory purposes: this concerns the medical device sector and specifies the requirements for the quality management system of organizations operating both in the design and production of medical devices, and in the design and provision of related services.EN ISO 14971 Medical devices—Application of risk management to medical devices: this specifies terminology, principles and a process for risk management of medical devices, including software as a medical device and in vitro diagnostic medical devices.MDCG 2019-16 Guidance in Cybersecurity for medical devices: this is intended to provide medical device manufacturers with guidelines for meeting essential hardware requirements, IT network characteristics and IT security measures, including protection against unauthorized access.

Considering the risk of possible harm to patients or other users as a result of dangerous situations resulting from the use of software, MDSW are divided into three classes:Class A: includes software that does not contribute to the occurrence of dangerous situations or, in any case, the risk control measures external to the software (e.g., health procedures) are sufficient to guarantee an acceptable risk.Class B: includes software that contributes to the occurrence of dangerous situations for which risk control measures external to the software are not sufficient to guarantee an acceptable risk and the possible harm is not serious.Class C: includes software that contributes to the occurrence of dangerous situations for which risk control measures external to the software are not sufficient to guarantee an acceptable risk and the possible harm is severe.

The SIMpLE software system has not been designed for diagnostic purposes, but it is intended for telemonitoring and control. It, therefore, falls into class A because a software malfunction does not cause a dangerous situation for the patients. The SIMpLE sensor nodes are commercial sensors. They are hardware devices compliant with ISO 13485. There is no risk of electric shock as they are low current devices and are battery-powered.

## Figures and Tables

**Figure 1 sensors-21-07239-f001:**
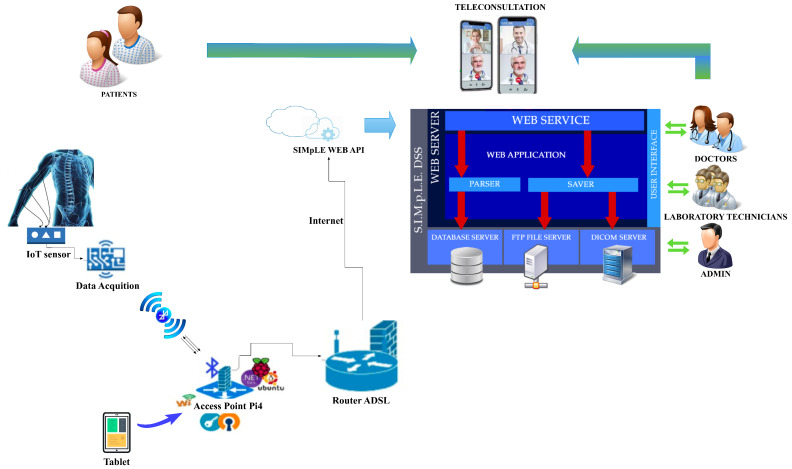
SIMpLE system overview.

**Figure 2 sensors-21-07239-f002:**
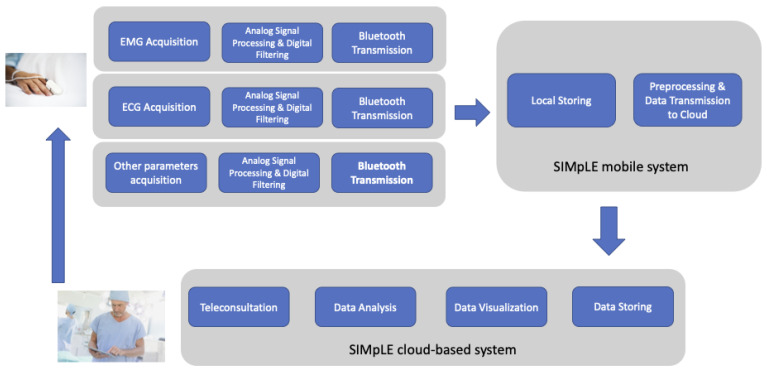
The SIMpLE system general architecture which consists of a data acquisition module (commercial sensors), a mobile system and a cloud-based system.

**Figure 3 sensors-21-07239-f003:**
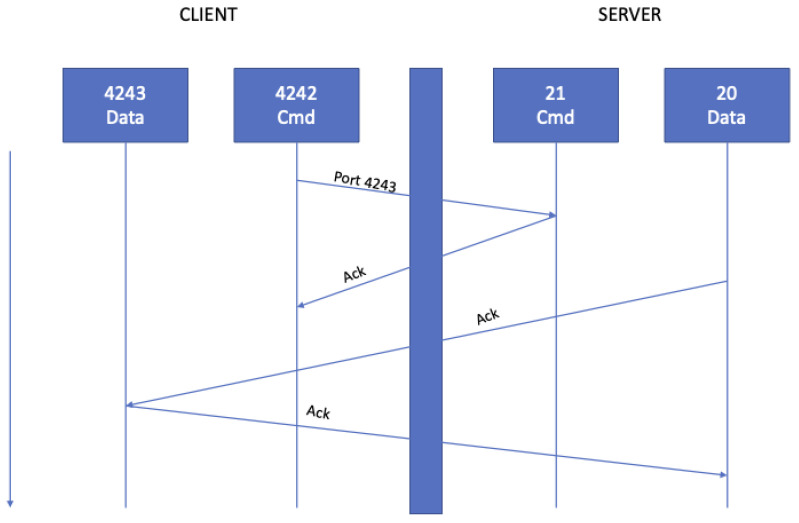
TCP/IP commands scheme.

**Figure 4 sensors-21-07239-f004:**
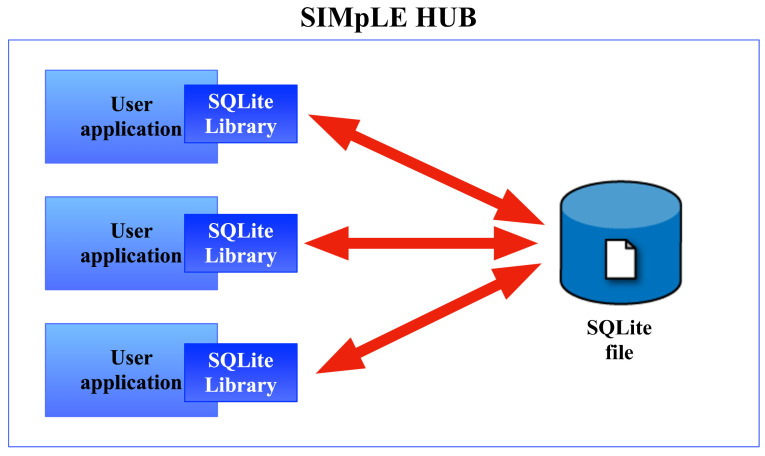
SQLite serverless architecture.

**Figure 5 sensors-21-07239-f005:**
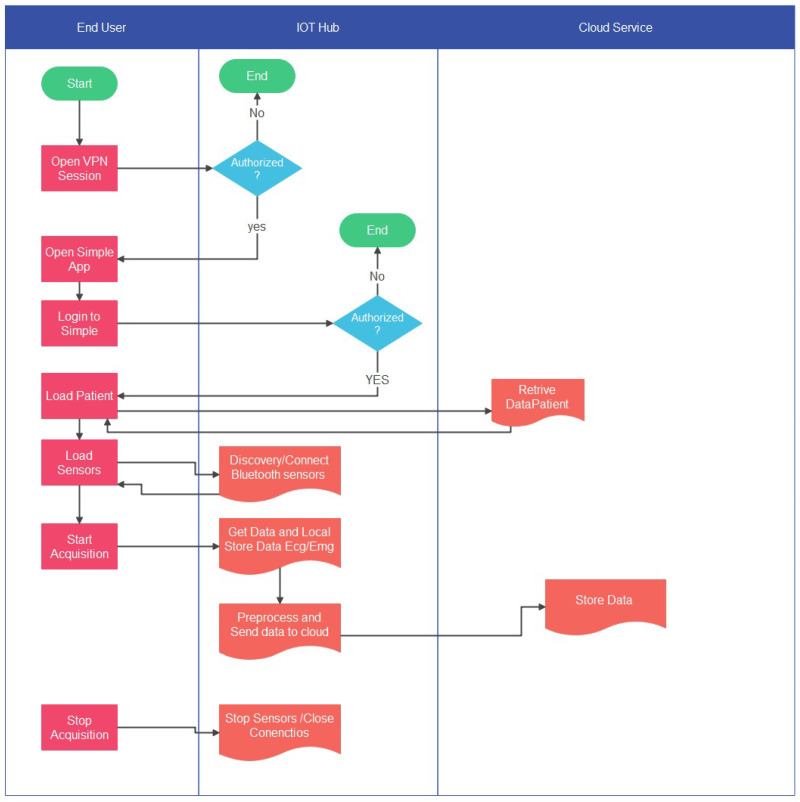
Flow chart relative to the communication between SIMpLE modules.

**Figure 6 sensors-21-07239-f006:**
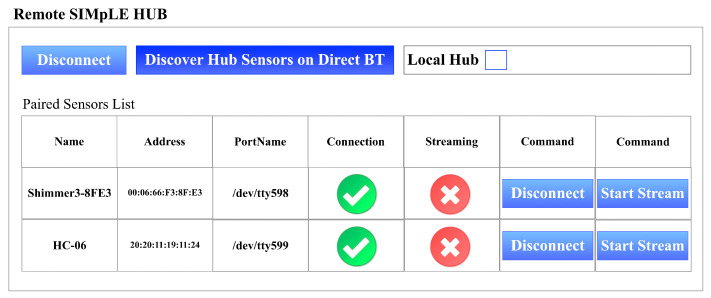
Sensors connected to the Remote SIMpLE HUB.

**Figure 7 sensors-21-07239-f007:**
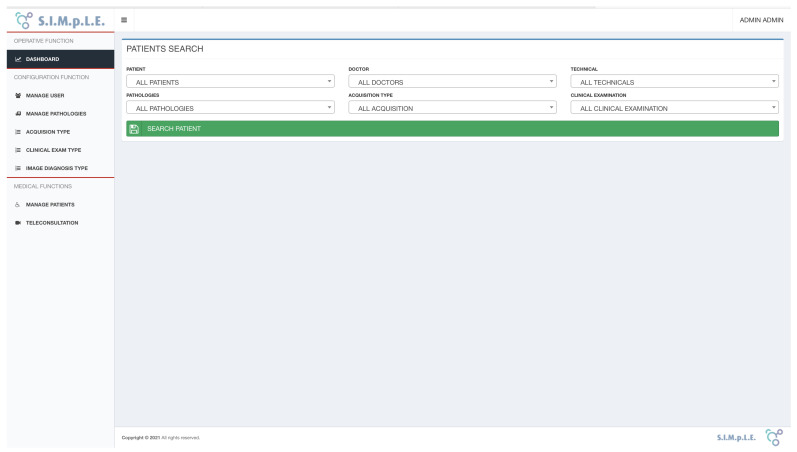
SIMpLE operational dashboard.

**Figure 8 sensors-21-07239-f008:**
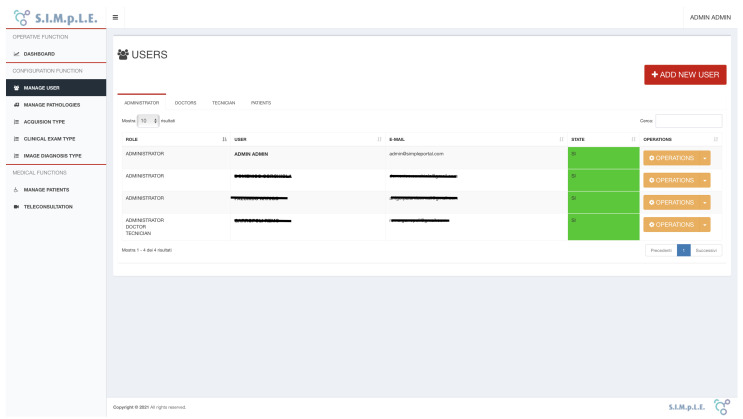
Example of the functionality of viewing, inserting, modifying and deleting users. Users can be Administrators, Doctors, Technicians and Patients and are always managed from the same interface.

**Figure 9 sensors-21-07239-f009:**
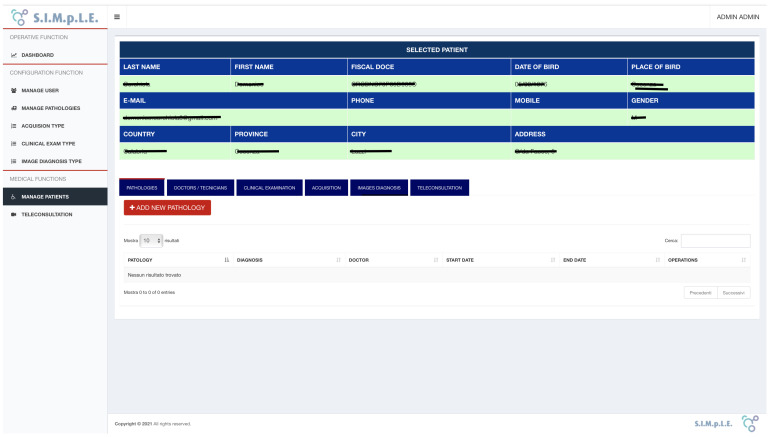
The Patient summary.

**Figure 10 sensors-21-07239-f010:**
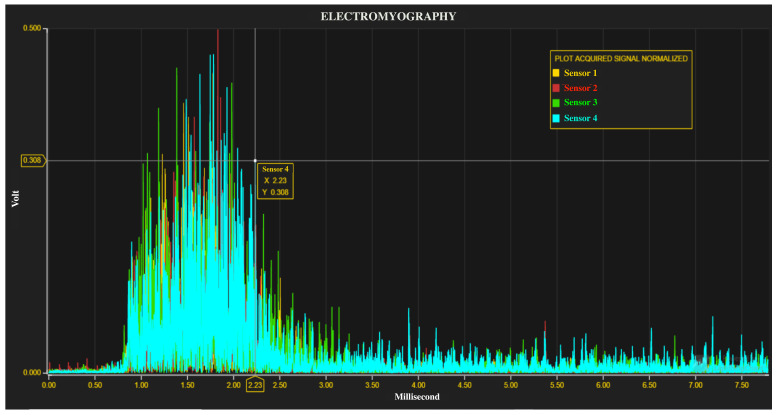
Web viewer for biopotential signals: an example of rectified EMG signals.

**Figure 11 sensors-21-07239-f011:**
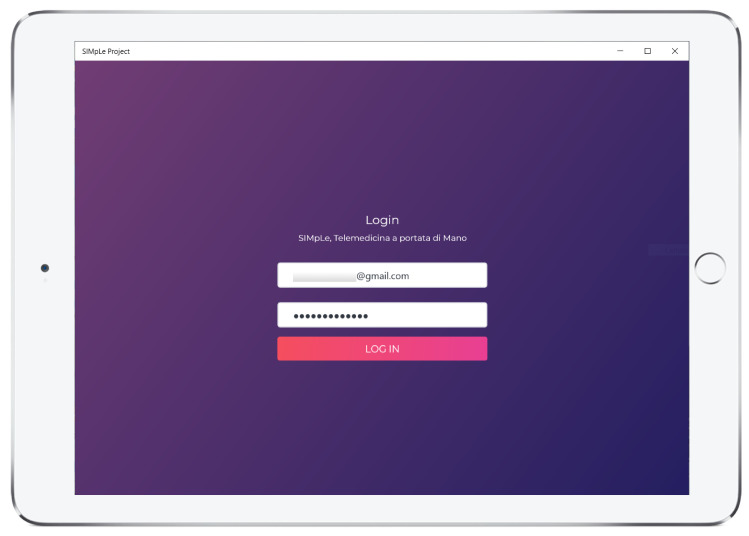
Login view for Windows, Adroid and IOS.

**Figure 12 sensors-21-07239-f012:**
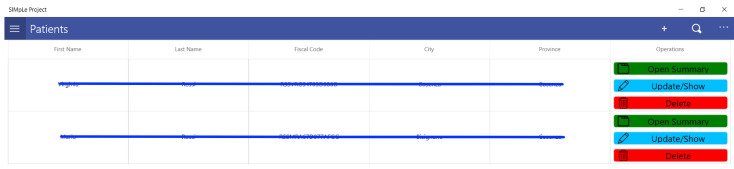
Patients list view.

**Figure 13 sensors-21-07239-f013:**
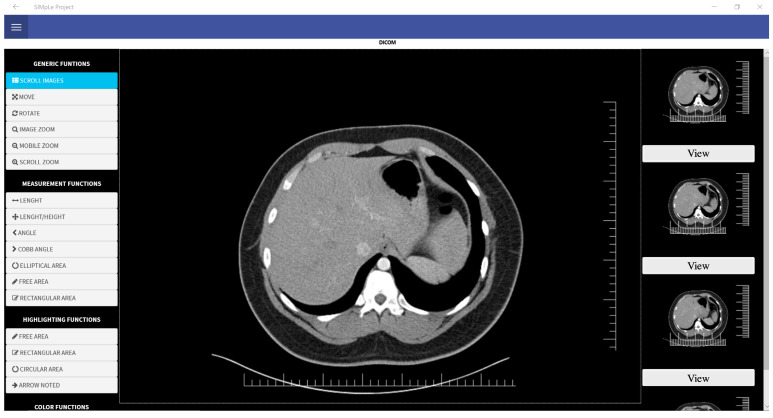
DICOM Viewer.

## Data Availability

The data presented in this study are available on request from the corresponding author. The data are not publicly available due to privacy.

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
