# Peer review of "SIMpLE: A Mobile Cloud-Based System for Health Monitoring of People with ALSâ€"

_sensors, 2021, doi:10.3390/s21217239_

Round 1
Reviewer 1 Report
This article presented the development of the mobile cloud-based system for health monitoring systems for ALS people. The platform allows medical staff to monitor patients' health conditions at a distance, allowing them to intervene directly on the electronic instrumentation in remote mode. The developed system seems interesting to me. However, the paper quality from the reader's perspective is hard to read.
Below are my suggestions for the improvement of the paper's overall quality and its structures.
- The author should mention the novelty of the developed system and describe how it is different from existing systems in terms of performance or other factors.
- The structures of the paper are not in compliance with journal format; the authors are requested to restructures as follows:
- Introduction
- Method and Materials or Methodology
- First, the author need to briefly describe the overall system in one workflow diagram, which has two central applications (i.e., a client application for data collection from a patient in real-time and a web application for physicians for real-time monitoring)
- SIMpLE: a mobile Application
- design and implementation
- SIMpLE: Web Application cloud-based system for health monitoring
- design and implementation
- SIMpLE: a mobile Application
- The authors added unnecessary details in both application designs, such as DOSS, VPN, Servers. Instead of mentioning these details, It will be worth mentioning how individual system components work together in Algorithms.
- The system should be validated in terms of usability or User experience. For User experience evaluation, both patients and physicians must check the user perception towards the developed system.
Reviewer 2 Report
The paper focuses on the adoption of telemonitoring services during the pandemic for people affected by chronic diseases such as the Amyotrophic Lateral Sclerosis (ALS).
Remote monitoring and teleconsulting are surely essential services for fragile patients such as those with the ALS, especially during the COVID19 pandemy, but I would say even in less dangerous times.
Authors present an innovative secure medical monitoring and teleconsultation mobile cloud-based system for disabled people. The infrastructure is able to acquire and process vital signs and especially concerning ECG and EMG.
Medical staff is provided with functionalities to monitor patient’s conditions remotely and to intervene directly on the electronic instrumentation in case of need.
The paper is well written and the solution is interesting. Moreover, the article is certainly of interest for the readers of this journal.
I have only one main concernment that I would like to see addressed by the authors.
The system may fall under the regulatory framework for Medical Devices and Software as a Medical Device.
It is certainly a prototype obtained as a result of research activities, but this will not free the clinical staff from responsibilities in case of system failures.
I guess that this problem must be clearly introduced within the paper explaining that an engegnerised version of the prototype should be certified as compliant to the MD European regulations.
To conclude, I suggest the authors to add a paragraph that introduces the problem and refers to:
- The European medical device regulation (https://eur-lex.europa.eu/legal-content/EN/TXT/PDF/?uri=CELEX:32017R0745)
- The ISO 13485 quality standard for medical devices (https://www.iso.org/standard/59752.html)
- A methodology for risk assessment in Software as Medical Devices (https://ieeexplore.ieee.org/abstract/document/9194991)
- The ISO 14971 standard for risk management ( https://www.iso.org/standard/72704.html)
Round 2
Reviewer 1 Report
The authors have addressed all of my comments and concerns in the revised version.